# Localized Additive Explanations (LAX)

## Abstract

We propose a novel technique for training an interpretable artificial neural network for image classification. Models trained with our method generate their own "visual explanation" for a prediction during the prediction step. Our approach, Localized Additive eXplanations (LAX), trains a neural network to predict the areas of an image, which, if occluded (dropped out), would prevent the learner from predicting the correct class. These regions must be essential to the prediction mechanism modeled by the learner, thus extracting regions that contain essential signals used for prediction. This method is applicable to any image classification problem, but requires training a model with a special structure that can be used to produce such visualizations. By utilizing this custom structure we bypass the need for an external explanation step, achieve higher granularity of evidence localization, and a performance benefit by incorporating the explanation generation in the prediction step. Our approach sacrifices a small amount of classification accuracy for the benefit of reduced time-to-explanation. LAX models are trained to optimize objectives which ensure that the signals identified by the explanation are 'correct' (they are evidence for a class), 'complete' (essential evidence for predicting a class), and 'exclusive' (not identifying anything other than evidence for a class). When the joint objective is achieved well, the resultant neural network is an accurate, but more importantly the same network provides a truthful explanation along with each prediction.

## 1 Introduction

Machine learning algorithms' predictions have been widely used either directly or to inform human experts for downstream tasks; e.g., predicting recidivism (Dressel & Farid, 2018), deciding on college admissions (D'Amour et al., 2020), and others. In this context, it becomes essential to provide *explanations* for machine learning models' predictions. Hence, an important pillar for *trustworthy machine learning* is that of *explainability*; a field that falls under the broader umbrella of *eXplainable Artificial Intelligence (XAI)* (Arrieta et al., 2020; Das & Rad, 2020; Burkart & Huber, 2021). Explanations provide benefits including, but not limited to, promoting *trust*, allowing for *accountability* of algorithms, and *causality* that highlights input-output relationships. In this context, unsurprisingly, a lot of effort is devoted to create methods whereby predictions of machine learning models can be explained.

When the learnt models are not interpretable by design, providing explanations can be a challenge. For example, linear combinations of features have been used widely, from perceptrons to logistic and linear regression. It is easy to understand such weighted sums and the influence of individual attributes to the prediction. However, when many such modules are combined to form artificial neural networks (ANNs), the input attributes contribute in delicate, difficult to express, ways to the final output. Hence, there is a trade-off between performant models and explainability (Rai, 2020). Our work seeks to improve on prior work providing explanations to classification tasks made by ANNs.

**Summary of Our Contribution**   Typical ANNs for image classification leverage convolutional or attention mechanisms to extract features by interleaving these mechanisms with downsampling steps and eventually producing a probability vector. We propose a fundamentally different way of solving the image classification task by forcing the model to produce an output of the same spatial dimension of the image. This output,

which attempts to allocate probability mass for a class in the prediction in the same spatial location as the evidence for that class, both serves as a prediction and is used to directly compute the output probability vector.

A Localized Additive Explanations (LAX) ANN has three outputs used to assess its performance: 1. the probability vector output of a traditional ANN for classification, 2. the output when the evidence the model identifies the predicted class of the image has been masked, and 3. the output when we mask all the model-identified class-relevant pixels. The first output is subject to supervision using the class labels. The other two outputs are designed to assess the ability of the model to segment the evidence it used to make its classification. Fundamentally we do not have ground-truth labels for the masks, so the other two model outputs are used to achieve a good mask in a self-supervised manner. To identify class-relevant pixels the model effectively generates an *"occlusion mask"* during its prediction step which serves as an explanation of the prediction. This explanation can be verified for correctness using only model outputs, and requires no external evaluation of the model. By incorporating the explanation in the prediction step, our method is hundreds of times faster than comparable XAI methods.

Our method, Localized Additive eXplanations (LAX), provides explanations that are *"localized,"* i.e., identifying the position in the image used for predicting a class, and *"additive,"* e.g., in the examples shown in Figure 1 the area of the prediction mask belonging to a certain color is directly proportional to the probability of that class.

## 2    Related Work

Modern ANNs are used with increasing frequency in various applications due to their predictive accuracy. However, because the prediction rules of ANNs are very complicated and hard to express, proxy methods are needed for their explainability. Such methods can be general, model-agnostic, XAI methods that apply to any model space and thus to ANNs as well, or XAI methods that are tailored to specific model classes.

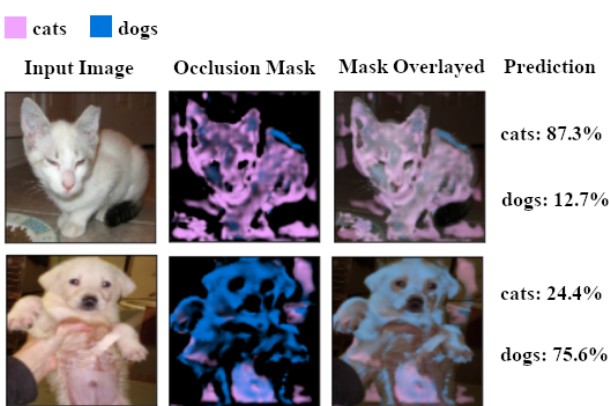

Figure 1: Examples from the Cats and Dogs dataset. The explanation of our model's prediction is shown in the middle column. The last column shows our explanation overlaid on top of the input image. We also see the probability assigned by the model to each one of the two classes.

**Model-Agnostic XAI Methods**  Local Interpretable Model-Agnostic Explanations *(LIME)* generates a "local" explanation by training an interpretable model (e.g., LASSO regression or decision tree) using input-output pairs generated by the model in the neighborhood of an instance to be explained (Ribeiro et al., 2016). Note that the "local" in LIME refers to locality in the input space to the model, not locality within an image example. When classifying images, each image is divided into a set of disjoint contiguous regions, called *superpixels*. The LIME explanation is then the most influential superpixels in the input, and optionally the order of their importance. One line of work investigates metrics for calculating the influence of inputs (or features) to the predicted outputs (Datta et al., 2016; Bhatt et al., 2020). Lundberg & Lee (2017) use the game-theoretic notion of the Shapley value, to provide Shapley Additive Explanations *(SHAP)*. Exact calculation of the Shapley value is costly, including several natural cases (den Broeck et al., 2022). In fact, issues arise beyond computational complexity (Kumar et al., 2020). Shapley values for image classification are approximated by evaluating the effect of removing a square region in the image on the classification probabilities. Methods such as LIME and SHAP can be vulnerable to adversarial inputs (Slack et al., 2020), while Slack et al. (2021) introduce a *Bayesian* version that attaches uncertainty levels to the provided explanations. Additional model agnostic methods include *Anchors* (Ribeiro et al., 2018), *MAPLE* (Plumb et al., 2018), and *prototypes* (Chen et al., 2019).

**Rate Distortion XAI Methods** Rate Distortion Explanations, first proposed in MacDonald et al. (2019), attempt to find the minimal area in an input image responsible for the model's prediction. This is achieved by finding the maximum mask by area that can be applied to an input without altering the predicted class. Several RDE methods have been proposed in prior work such as CartoonX Kolek et al. (2022), WaveletX Kolek et al. (2023), and ShearletX Kolek et al. (2023). These methods rely on optimizing a mask for each prediction for which one would like an explanation. As a consequence, we find these methods interesting but too costly to be practical in scenarios where an explanation is required for every prediction.

**ANN-Specific XAI Methods** Zeiler & Fergus (2014) present the first and possibly simplest method of visualizing the saliency of an input feature using their novel *"deconvnet"* approach, where the idea is to identify the contribution of each feature in a CNN, by following backwards the computation that was performed and allowed the prediction in the first place. The paper also presents another very simple method of evaluation wherein a gray patch is tiled across the image to occlude parts of the image and the change in the prediction is measured. We will refer to this method as 'Occlusion Sensitivity.' Simonyan et al. (2014) leverage the gradient of the probability of predicting a class with respect to each input feature, leading to *saliency maps*. Roughly, saliency maps provide a gradient heatmap over the input pixels that justifies which features to which classification is most sensitive locally; see also (Kim et al., 2022; Wagner et al., 2019). Other examples include Layer-wise Relevance Propagation (*LRP*, Bach et al., 2015), Class Activation Mapping *(CAM)* and the closely related methods of *GRAD-CAM* and *Guided GRAD-CAM* (Selvaraju et al., 2020), *interpretable CNNs* (Zhang et al., 2018), *neural additive models (NAMs)* (Agarwal et al., 2021), *B-cos Networks* (Böhle et al., 2022), and D-RISE for object detection (Petsiuk et al., 2021), to name a few. Another direction of exploiting gradients is with the method of *integrated gradients* (Sundararajan et al., 2017). Shrikumar et al. (2017) in *DeepLIFT*, explain the prediction of a particular input with respect to a reference input, by looking at the difference between these two inputs. Kim et al. (2018) explain the internal state of an ANN in terms of human-friendly *concepts*, Dhurandhar et al. (2018) propose *contrastive explanations*, while Brendel & Bethge (2019) propose the use of *bags of local features*.

## 3 Notation

For two vectors $\boldsymbol{a}, \boldsymbol{b} \in \mathbb{R}^n$, we denote their inner product $\langle \boldsymbol{a}, \boldsymbol{b} \rangle = \sum_{i=1}^n a_i b_i$. We use $\mathcal{X}$ to denote the set of instances and $\mathcal{Y}$ to denote the set of labels. Let $k = |\mathcal{Y}|$. As our applications will be on images, an instance $x \in \mathcal{X}$ is a tensor in $\mathbb{R}^{W \times H \times ch}$, where $W$ is the width of the image, $H$ is the height, and $ch$ is the number of channels ($ch = 3$ for red, green, blue values in all cases). For two tensors $A, B$ that have the same dimensions, we define $A \odot B$ to be their Hadamard product, that is, element-wise multiplication of the two tensors. We use $\mathcal{F}$ to denote the *model space* from where a learnt model is selected by a learning algorithm. While traditionally we define models to be of the form $f \colon \mathcal{X} \to \mathcal{Y}$, for our purposes we require knowledge of the full output measure of real numbers (probabilities) that are associated with the different classes, and hence it is convenient to define learnt models to be of the form $f \colon \mathcal{X} \to \mathbb{R}^k$. This approach also simplifies the presentation of Section 4 about the architecture that we use. Hence, a label $y \in \mathcal{Y}$ is truly of the form $\boldsymbol{y} \in \mathbb{R}^k$, where for every different class we use a different one-hot-encoding. In this light, loss functions have the form $\ell \colon \mathbb{R}^k \times \mathbb{R}^k \to \mathbb{R}_{\geq 0}$. For a sample $S$ of images $((x_1, \boldsymbol{y}_1), \ldots, (x_m, \boldsymbol{y}_m))$ we define the *mean image* $\mu \in \mathbb{R}^{W \times H \times ch}$ of $S$, where $\mu_{i,j,k} = \frac{1}{m} \sum_{r=1}^m (x_r)_{i,j,k}$.

## 4 Method

An overview of our method is show in Figure 2. LAX utilizes an image-to-image model, which makes three predictions. The image-to-image model in the figure is labeled 'U-net' referring to the style of image-to-image model presented by Ronneberger et al. (2015), but any sufficiently powerful image-to-image translation model could be used. In our experiments we utilize a neural network that uses 'skip' connections inspired by the 'U-net' but uses blocks similar to those in 'Focal Modulation Networks' (Yang et al., 2022) for higher resolutions and 'Multi-Headed Self-Attention' (Vaswani et al., 2017) modules at lower resolution. See our code for further implementation details. The input to the image-to-image model is in all cases an image tensor of size $W \times H \times ch$ where $W, H, ch$ as defined in Section 3. The output in all cases is a mask tensor of

size $W \times H \times (k+1)$. When the model achieves its objective well, slices of size one along the last axis of the output tensor correspond to masks, which identify pixels that, if dropped out, would result in the model not being able to predict a certain class. We will refer to these slices of the output volume as *occlusion masks*. The last output occlusion mask identifies pixels that contribute to no class and are never dropped out. The probability vectors $p_0, p_1,$ and $p_2$, shown in Figure 2, are explained below.

## 4.1 Training

One training step of the model requires three inferences of the image-to-image network. Each of these inferences produces an output that incurs a loss. We describe a training step in three parts below.

**Part I** Initially the model takes as input the image for classification and predicts the parts of the image, were they to be removed, would result in a decrease of the probability of a certain class for all classes. This follows because the same model parameters are used in each step, and in subsequent steps the model will be encouraged by the loss function to predict a region such that if it were removed the model would perform worse. The softmax function is applied across the channel dimension of each pixel. The class prediction is given by the normalized means of the channels in the output excluding the final 'no class' channel. The resulting probability vector $p_0$ is the first output of the model.

Figure 2: Model Architecture. Shown are three prediction steps of the model on a given image. In the first step the model is presented with the image, in the second it is presented with the image when the evidence for the dominant class is removed, and at the bottom we see the output when the evidence for all classes is removed.

**Part II** The second inference of the model takes as input the image for classification combined with the *mean image $\mu$* of the training set (see Section 3) such that the pixels in the mask corresponding to the predicted class from the previous step are dropped out. Formally, given $\mu$, the occlusion mask $M_{pred} \in \mathbb{R}^{W \times H}$ corresponding to the predicted class, and $x \in \mathbb{R}^{W \times H \times ch}$ the input image, the operation is:

$$x' = (M_{pred} \odot \mu) + ((1 - M_{pred}) \odot x), \tag{1}$$

where the Hadamard product $\odot$ is broadcast where applicable. This image $x'$ is input to the same image-to-image model as in the first step, which produces $k+1$ output occlusion masks. These occlusion masks are averaged, and the averages of the class occlusion masks are again renormalized to produce a probability vector output. This probability vector $p_1$ is the second output of the model.

**Part III** The third inference of the model takes as input the image for classification combined with the mean of the training set such that all the pixels in the mask corresponding to any class from the first step are dropped out. Formally, given mask $M \in \mathbb{R}^{W \times H \times k+1}$ and $M_{sum} = \sum_{i=0}^{k} M_i$ we have

$$x'' = (M_{sum} \odot \mu) + ((1 - M_{sum}) \odot x). \tag{2}$$

This image $x''$ is input to the same image-to-image model as in the first step which produces $k+1$ output occlusion masks. These occlusion masks are averaged, and the averages of the class occlusion masks are again renormalized to produce a probability vector output. This probability vector $p_2$ is the third output of the model.

Each of the outputs from the steps above incurs a loss. The first output $p_0$ incurs cross-entropy loss $\ell_0(y, p_0)$ based on its prediction compared to the true label of the instance. The second output $p_1$ incurs a loss $\ell_1(p_0, p_1)$

based on the similarity of $p_1$ and $p_0$. First both vectors are renormalized to have a Euclidean vector norm of 1. We will call these renormalized vectors $p_0'$ and $p_1'$. Then the model suffers loss $\ell_1(p_0, p_1) = -\ln\left(1 - \langle p_0', p_1' \rangle\right)$. The last output $p_2$ incurs a loss $\ell_2(p_2) = -\ln\left(\overline{p_2}/p_2^*\right)$, where $\overline{p_2}$ is the mean of the $p_{2,i}$'s, and $p_2^*$ is the max of the $p_{2,i}$'s.

All the above yield the following definition.

**Definition 1** (LAX Loss). *For a given model $f$ that is equipped with the LAX architecture and an input labeled image $(x, \boldsymbol{y})$, upon prediction, the model suffers loss:*

$$\mathcal{L}_{LAX}(f, x, \boldsymbol{y}) = \ell_0(\underbrace{f(x)}_{\boldsymbol{p_0}}, \boldsymbol{y}) + \alpha \ell_1(\underbrace{f(x)}_{\boldsymbol{p_0}}, \underbrace{f(x')}_{\boldsymbol{p_1}}) + \beta \ell_2(\underbrace{f(x'')}_{\boldsymbol{p_2}}), \tag{3}$$

*where $\alpha$ and $\beta$ are hyperparameters, and $x'$ and $x''$ are obtained from (1) and (2) respectively.*

**Intuition**   Intuitively, $\ell_0$ encourages the model to mask the evidence essential to predicting a class, with the occlusion mask corresponding to the class by penalizing the model for adding probability mass to the incorrect classes' occlusion masks. In sequence, $\ell_1$ encourages the model to mask nothing but the evidence essential for the predicted class, as this loss is the lowest when $p_1$ is orthogonal to $p_0$. Finally, $\ell_2$ encourages the model to mask all of the evidence essential for any class, as the model suffers no loss for using the final occlusion mask belonging to no class, except when it would otherwise be able to predict any class using those pixels. Perhaps more intuitively, this loss encourages the model to predict uniformly and with maximal uncertainty after masking (e.g. the max and mean of the probability vectors are equal to $\frac{1}{k}$). So, all together, the model's objective can be understood as masking the evidence it is using within the input, all of the evidence essential for any class, and nothing but the evidence. Note that we qualify *evidence* with *essential* as the masks need only remove essential signals for the objective function to be optimized. We would be quite fortunate if the model effectively segmented the entire object belonging to a class, and indeed this is one possible optimal solution, but it is not necessary to achieve minimum loss.

**Naming**   The occlusion masks generated by the model identify spatial regions that contain signals essential to classification by design, because when they are removed the same model can no longer predict the correct class, and therefore we call these explanations "localized". Note that this does not imply the pixels are logically related to the ground truth, simply that the model is "truthful" in identifying which pixels it is using to predict the predicted class. Furthermore, explanations are "additive" because the class probabilities are directly computed from the occlusion masks by taking the mean value of the mask for each class divided by the sum of the mean values for all masks. Hence, the number of pixels covered by a mask for a class is directly proportional to the probability of that class. In the examples shown, the area of the prediction mask belonging to a certain color is directly proportional to the probability of that class.

### 4.2   Inference

**Definition 2** (Accuracy). *Given a sample $S = ((x_1, \boldsymbol{y_1}), \ldots, (x_m, \boldsymbol{y_m}))$ and a LAX model $f$ predicting $\left(\left(\boldsymbol{p_0^{(1)}}, \boldsymbol{p_1^{(1)}}, \boldsymbol{p_2^{(1)}}\right), \ldots, \left(\boldsymbol{p_0^{(m)}}, \boldsymbol{p_1^{(m)}}, \boldsymbol{p_2^{(m)}}\right)\right)$ on $S$, $f$ has accuracies in the three components $j \in \{0, 1, 2\}$:*

$$A_j^{(f)} = 1 - \frac{1}{m} \sum_{i=1}^{m} \mathbf{1} \left\{ \underset{1 \leq q \leq k}{\arg\max} \left( p_j^{(i)} \right)_q \cap \underset{1 \leq r \leq k}{\arg\max} (y_i)_r = \emptyset \right\},$$

*where $\mathbf{1}\{\mathcal{E}\} = 1$ if event $\mathcal{E}$ holds; otherwise, $\mathbf{1}\{\mathcal{E}\} = 0$.*

During training we encourage a model $f$ to achieve its objectives well outside of simply prediction accuracy. If this is the case and $f$ generalizes well to unseen data, then we might reasonably be sure that on a given instance it is indeed explaining its prediction satisfactorily. We can measure how well $f$ achieves these objectives by observing the accuracies $A_0^{(f)}$, $A_1^{(f)}$, and $A_2^{(f)}$, corresponding to the outputs $\boldsymbol{p_0}$, $\boldsymbol{p_1}$, and $\boldsymbol{p_2}$. Ideally, we should have $A_0^{(f)} = 100\%$, $A_1^{(f)} = 0\%$, and $A_2^{(f)} = 1/k$. If we are happy to accept that its predictions, as well as its explanations, are likely reasonable based on test set performance, then we can

discard all of the masking steps and simply use the first inference of the image-to-image model. In practice this is desirable as the computational cost of inference with explanation is low. However, we are not at the mercy of aggregate performance statistics when evaluating the quality of a particular explanation. If we retain all three of the prediction steps during inference, we can measure how the predicted label changes between $p_0$, $p_1$, and $p_2$. If the argmax of $p_0$ and $p_1$ differ and $p_2$ has a lower maximum value than $p_0$, then we know that the model has achieved its secondary objectives well on that instance. We revisit this discussion in Section 5 but we are now ready to define the *model of best occlusion.*

**Definition 3** (Occlusion Quality)**.** *For a LAX model $f$, with output accuracies $A_0^{(f)}, A_1^{(f)}$, and $A_2^{(f)}$, we define the* occlusion quality *to be $Q_{LAX}(f) = A_0^{(f)} - \left| A_2^{(f)} - 1/k \right|$.*

**Definition 4** (Model of Best Occlusion)**.** *Let $\mathcal{F}_{LAX}$ be a set of different LAX models. We define the* model of best occlusion $f^* \in \mathcal{F}_{LAX}$ *to be $f^* \in \arg\max_{f \in \mathcal{F}_{LAX}} Q_{LAX}(f)$.*

### 4.3 Trainable Components

The only trainable portion of this architecture is the single image-to-image model labeled 'U-net' in Figure 2. Each U-net in the diagram is identical, i.e., they all share the same weights. During training this is advantageous as the model weights can be stored only once. During the backward pass, the weights of the U-net in our experiments are updated five times. The first time is a consequence of the loss suffered due to $\ell_0$. Then the loss suffered from $\ell_1$ and $\ell_2$ are propagated through the entire path through the execution graph from output to the input image; thus, twice each through the weights of the U-net. The propagation of the gradient is *not stopped* after the nearest application of the U-net to the output, as we found this empirically a more successful approach than the alternative in preliminary investigations. Further analysis is necessary to determine the effect of that potential stop.

**Remark 1** (On Object Detection and Localization)**.** *Within computer vision one may aim to create bounding boxes (Everingham et al., 2015) or semantic segments (Shelhamer et al., 2017) within images, identifying specific objects of interest. In particular, with* weakly supervised object localization (WSOL) *one identifies one or more instances of a particular object in an image by using only an image-level label. While WSOL has gained popularity, it is an ill-posed task and realizing improvements over prior work can be a delicate process (Choe et al., 2020). Despite the fact that this problem is similar to many explanation methods for image classification, our work is not trying to create bounding boxes or semantic segments for specific objects within an image. We propose an architecture that partitions the pixels of a given image based on the pixels' contribution for the predicted class probabilities regardless of their ground truth relevance.*

## 5 Experimental Evaluation

### 5.1 Setup for Experiments

We evaluate our models on a three-way split: 70% of the data is used for training, 10% is reserved for validation, and 20% is reserved for testing. Validation data was used to tune both the model's hyperparameters as well as the hyperparameters associated with data augmentation and training. As with many deep learning experiments the choice of value for the batch size, the choice of an optimizer, and the choice of data augmentation strategy are non-trivial. Details of our training hyperparameters can be found in Appendix A and the code we have included as a part of our supplementary material. Testing data is evaluated only once hyperparameter selection has completed for each dataset. Only testing and validation set images are displayed in the figures.

**Model Optimization**   We use three loss functions for our model optimization, but all of our objectives can be seen as surrogate losses for our true objective. Our true objective is for the model to mask the evidence it uses for prediction, and to predict correctly. We measure the fitness of a model to this objective by how accurate the model is when presented with the original input, and how accurate the model is when presented with an image for which it has masked out all of the class evidence. In the first case we would naturally like our model to be maximally accurate, and in the second case we would like our model to guess randomly.

**Baselines** We compare our model to two different models and five total comparable XAI methods for explaining image classification predictions. The first model we train is the Interpretable CNN (Zhang et al., 2018) which is an 'ante hoc' explanation method, and the second is the DenseNet model (Huang et al., 2017). We apply LIME, SHAP, Gradient (Saliency Map), and Occlusion Sensitivity methods as 'post hoc' explanation baselines. We build the 'interpretable' layers of the Interpretable CNN on top of DenseNet121 for the sake of a fair comparison between the ante hoc Interpretable CNN baseline and the post hoc baselines. For the baseline models we train each model at values of dropout in 5% increments from 0% to 50% and select the model with the highest validation accuracy. Dropout is applied after each convolutional layer as in Huang et al. (2017).

**Hardware Resources** Models were trained on four Nvidia A100 40GB GPUs.

### 5.2 Dataset

We utilize the common Cats and Dogs (Parkhi et al., 2012) dataset to provide an understanding of how our method performs and produces visualizations. We employ this relatively small and easy dataset to demonstrate the intuitive

Table 1: Class frequencies for Cats and Dogs.

| class | train instances (%) | validation instances (%) | testing instances (%) |
|-------|-----------------|---------------------|-------------------|
| cats | 8195 (50.3%) | 1183 (50.8%) | 2280 (49.0%) |
| dogs | 8088 (49.7%) | 1144 (49.2%) | 2372 (51.0%) |

and unintuitive nature of individual explanations. Cats and dogs have a clear notion of contiguous "objectness." In this dataset signals that are discriminative for classification are localized in discrete blob-like structures. It is easy for us, as humans, to determine what parts of the image contain real evidence for its class from those parts with spurious co-variates. As we will see, our models, like others, cannot always easily determine these differences. We believe this highlights the need for explanations which are truthful rather than just intuitive.

**Class Composition** Table 1 provides the information on the number of examples that we have for each class.

### 5.3 Evaluation on the Cats and Dogs Dataset

The models achieving the best accuracy and the best $Q_{\mathrm{LAX}}$ were both found with $\alpha = \beta = 2^{-10}$. Table 2 shows accuracy results on the Cats and Dogs dataset (best accuracy and best occluded), the DenseNet baseline, and the Interpretable CNN baseline. Table 3 presents relevant confusion matrices. Namely, table 3b presents the confusion matrix that corresponds to the classification obtained after masking (i.e., using output $p_2$ for classification), where again this accuracy is achieved by setting $\alpha = \beta = 2^{-10}$ in (3) since this was the setup that had the highest accuracy in the Cats and Dogs task. Table 3c presents the confusion matrix that corresponds to accuracy of the model of best occlusion and is also achieved by setting $\alpha = \beta = 2^{-10}$ in (3). Finally, Table 3d resents the confusion matrix of the best occluding model obtained after masking (i.e., using output $p_2$ for classification).

Table 2: Evaluation results on the two datasets. The testing accuracies of the LAX models are compared to their accuracies after the pixels identified as relevant have been masked. Recall that the best accuracy LAX model is the one that has the highest accuracy using $p_0$ for prediction, while the best occluded LAX model is the one that maximizes $Q_{\mathrm{LAX}}$ (see Definition 4) and still uses $p_0$ for prediction. Hence, the accuracy after masking the relevant evidence is equivalent to the accuracy if $p_2$ were used for prediction by either model. For $p_2$ accuracy, closer to $\frac{1}{k}$ is better where applicable. DenseNet and Interpretable DenseNet models do not have a notion of $p_2$ accuracy, so only their test set accuracy is displayed.

| model description | testing accuracy | $p_2$ testing accuracy |
|-------------------|------------------|------------------------|
| LAX (best accuracy) | 88.18% | 60.73% |
| LAX (best occluded) | 84.89% | 56.58% |
| DenseNet | 89.77% | n/a |
| Interpretable DenseNet | **90.60%** | n/a |

By varying the loss weights our method presents a tradeoff between the accuracy of the model and the occlusion performance (which corresponds to the power of the explanation). In all cases LAX provides a simple method by which to determine whether the explanation is correct: If the probability of the predicted class in $p_1$ is much less than that in $p_0$ then the explanation is accurate. In this sense the explanation cannot be untruthful.

Table 3: Confusion matrices for the Cats and Dogs models. Table 3a presents the confusion matrix for the model that produces the highest validation ($p_0$) accuracy, and Table 3b presents the confusion matrix for the masked ($p_2$) accuracy of the same model. Table 3c presents the confusion matrix for the model with the best occlusions, and Table 3d presents the confusion matrix for the masked ($p_2$) accuracy of the same model.

(a) Confusion matrix for the Cats and Dogs model with the highest validation accuracy. Results obtained with $\alpha = \beta = 2^{-10}$.

|  | predicted label | |
| --- | --- | --- |
| true label | cats | dogs |
| cats | 1891 | 389 |
| dogs | 155 | 2217 |

(b) Confusion matrix for Cats and Dogs model with the highest validation accuracy if $p_2$ were used for prediction. Results obtained with $\alpha = \beta = 2^{-10}$.

|  | predicted label | |
| --- | --- | --- |
| true label | cats | dogs |
| cats | 2271 | 9 |
| dogs | 1818 | 554 |

(c) Confusion matrix for best occluding model trained on Cats and Dogs. Results obtained with $\alpha = \beta = 2^{-10}$.

|  | predicted label | |
| --- | --- | --- |
| true label | cats | dogs |
| cats | 1898 | 382 |
| dogs | 321 | 2051 |

(d) Confusion matrix for best occluding model trained on Cats and Dogs if $p_2$ were used for prediction. Results obtained with $\alpha = \beta = 2^{-10}$.

|  | predicted label | |
| --- | --- | --- |
| true label | cats | dogs |
| cats | 2269 | 11 |
| dogs | 2009 | 363 |

### 5.3.1 The Importance of Truthful Explanations

Having identified the possibility that explanations might be successful or unsuccessful one may reasonably ask whether such cases even arise. Here we show that not only do they arise, but some unsuccessful explanations appeal to our intuitions more than some successful ones. This highlights the need for an objective measure of the accuracy or truthfulness of an explanation.

Figure 3a presents an effective occlusion map, where we can see that by masking out the evidence, the probability of selecting a particular class changes enough to impact the classification. This is not true in Figure 3b where the occlusion map appears to be a satisfactory explanation, but it does not meaningfully effect the predicted probability. These examples highlight the usefulness of the additional forward passes (i.e., $\text{argmax}(p_0) = \text{argmax}(p_1)$) to determine the fitness of an explanation, as one could be persuaded to accept the second explanation, but it turns out not to correspond to an effective occlusion map.

Figures 4 and 5 provide additional examples with comparisons between LAX and other explainability methods in the Cats and Dogs dataset. In Figure 4 we can see three examples of occlusions that meaningfully capture the ground truth signals in the image, and change the predicted probability by several percentage points, but do not manage to change the class label. These occlusions appear to be satisfactory explanations, and do manage to impact the predicted probability, but are not quite successful as they do not change the predicted class.

In Figure 5 we can see three more Cats and Dogs examples illustrating more possible occlusion cases. Figure 5a is almost successful, but does not quite manage to change the predicted class. Figure 5b seems to segment well the ground truth signal, but deceptively manages to increase the probability of the predicted class. Figure 5c Seems to do a very poor job of segmenting the ground truth signal, but manages to drastically reduce the probability of the predicted class indicating that it is in fact a very good explanation and a very successful occlusion.

All of these cases illustrate that what a neural network learns does not necessarily align with what a human would understand. Identifying these potential faults is an important part of interpretability. LAX models naturally lend themselves to interpretation by providing a way to detect potential shortcomings.

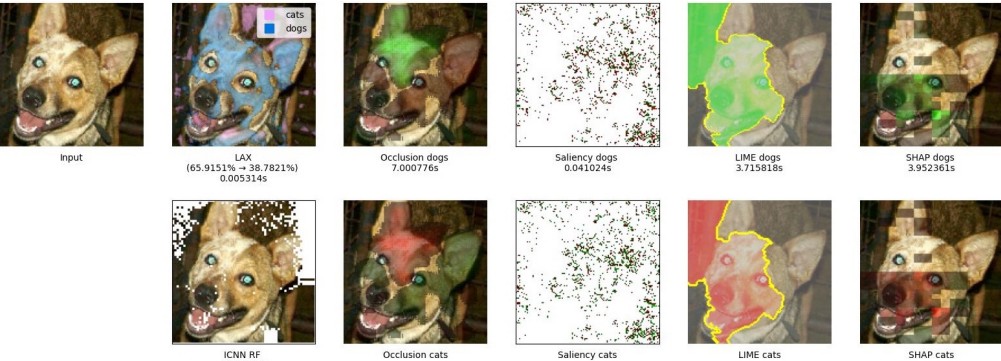

(a) A successful occlusion from the Cats and Dogs task. The model is predicting the correct label and the occlusion changes the predicted class. The numbers below 'LAX' on the figure represent the percent change in the predicted class (dogs) following the occlusion.

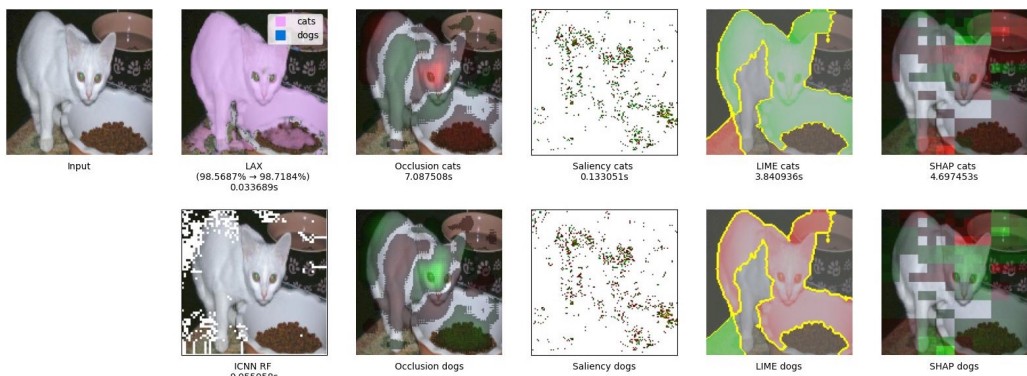

(b) An unsuccessful occlusion from the Cats and Dogs task. The model is predicting the correct label, but the occlusion doesn't significantly impact the predicted class probability. The numbers below 'LAX' on the figure represent the percent change in the predicted class (cats) following the occlusion.

Figure 3: Successful and unsuccessful occlusions from the Cats and Dogs task. Images are ordered top to bottom by their descending prediction probabilities. The first column contains the input image. The second column contains the output of the LAX model and the 'Receptive Field' of the Interpretable DenseNet. The last four columns are XAI methods (Occlusion Sensitivity, Saliency Maps, LIME, SHAP in that order) applied to DenseNet. Time in seconds to produce the explanation for the given instance is included for each method.

## 6 Discussion

**Advantages** Methods based on saliency maps suffer from: (I) highly overlapping information with low spatial frequency in the case of GRAD-CAM and Interpretable CNNs, or (II) very high spatial frequency information that highlights individual pixels in what frequently appears in a random/noisy way (Kim et al., 2019), or (III) thousands of inferences are needed in order to generate an explanation. Case I is undesirable because it is hard to see which features in the image are being used as discriminative evidence for a specific class, and it is hard to compare the explanations for different classes. Along similar lines, Rudin (2019) argues that "saliency does not explain anything except where the network is looking" providing a picture of a dog and saliency maps that look almost identical when predicting a dog or a musical instrument. While this can be a very valid point in prior work, in our approach such pathological cases cannot arise as we provide a unique saliency map where pixels are color-coded accordingly for the various classes. Case II is also undesirable because of the potentially fragmented explanations that do not take into account neighboring

Table 4: Compute time comparison for all methods at 128 by 128 resolution on a batch of size 32 examples. Times recorded on a machine with one Nvidia Ampere A100 GPU.

| method | forward passes per example | runtime (minutes) |
| --- | --- | --- |
| Saliency Maps | **1** | 0.028 |
| LAX (ours) | 3 | **0.002** |
| LIME | 2048 | 1.94 |
| SHAP | 2048 | 2.04 |
| Occlusion Sensitivity | 4096 | 3.36 |
| Interpretable CNN | 4096 | 4.70 |

pixels. While our approach, in principle, allows such pathological situations to arise, nevertheless, we did not observe any such case in our experiments. Case III corresponds to many available model-agnostic methods (SHAP, LIME, Interpretable CNNs, Occlusion Sensitivity) that require thousands of inferences to generate their explanation. These methods are computationally expensive to approximate, and none has an existing implementation for image classification that achieves high granularity and low latency. Our method is an order of magnitude faster than any of the baselines, and several orders of magnitude faster than some. Table 4 presents a runtime comparison of the explanation methods.[1] Our method is agnostic to the independence, location, or contiguity of essential signals (unlike SHAP, LIME, and Occlusion Sensitivity), and our method only relies on at most three different forward passes and provides the full and final explanation.

**Limitations**   Our method is architecture-dependent. Similar to Interpretable CNNs, the output of the ANN used for classification must have a special structure. The underlying image-to-image model can vary. Not unexpectedly our model suffers a mild performance detriment in terms of accuracy over the baselines when the occlusion quality is high. It is unclear if our method is applicable outside of image classification tasks. Finally, methods like GRAD-CAM and saliency maps provide an exact computation of an approximate explanation; e.g., they can exactly compute some property which approximately explains a prediction. Our method provides exact computation of a *probable* explanation.

## 7   Conclusion

We proposed a novel approach, Localized Additive eXplanations (LAX), for explaining image classification predictions. The approach relies on the application of three different losses that encourage the formation of occlusion masks that cover important features responsible for classification. Our experiments indicate that the proposed architecture performs well regardless of whether the explanations are expected to be contiguous segments of images or not. Moreover, the trained architecture provides explanation much faster compared to baselines, which could be critical for real-time applications, while at the same time providing qualitative results that are probably better than both methods. Finally, there are several ideas for future work, including the exploration of alternate methods for early stopping, computing the mean iteratively during training rather than using the batch mean, but perhaps the most important question is whether it is possible to output only one mask for all of the class evidence (or just the majority class) and this way escape the limitation of the output volume size in situations where we need to perform classification among many classes.

---

[1]We used the official SHAP and LIME implementations in python that are available at `https://shap.readthedocs.io/en/latest/` and `https://lime-ml.readthedocs.io/en/latest/` respectively.

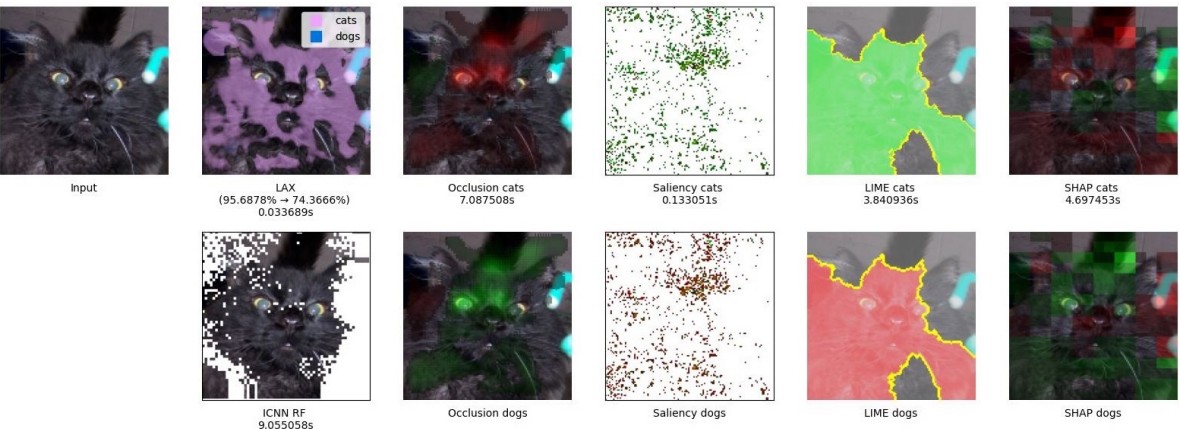

(a) A not-so-successful occlusion example that seems to identify the ground truth signal well.

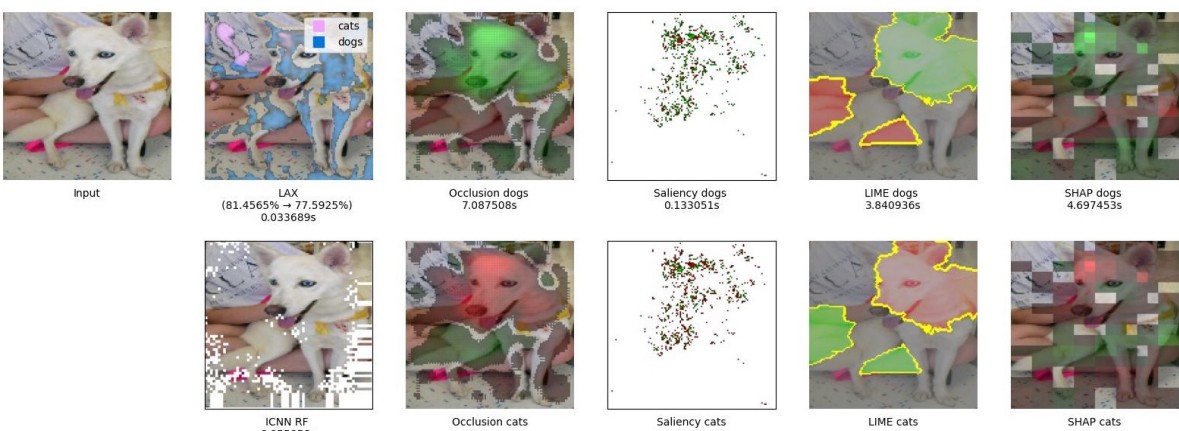

(b) A not-so-successful occlusion example that seems to identify the ground truth signal well.

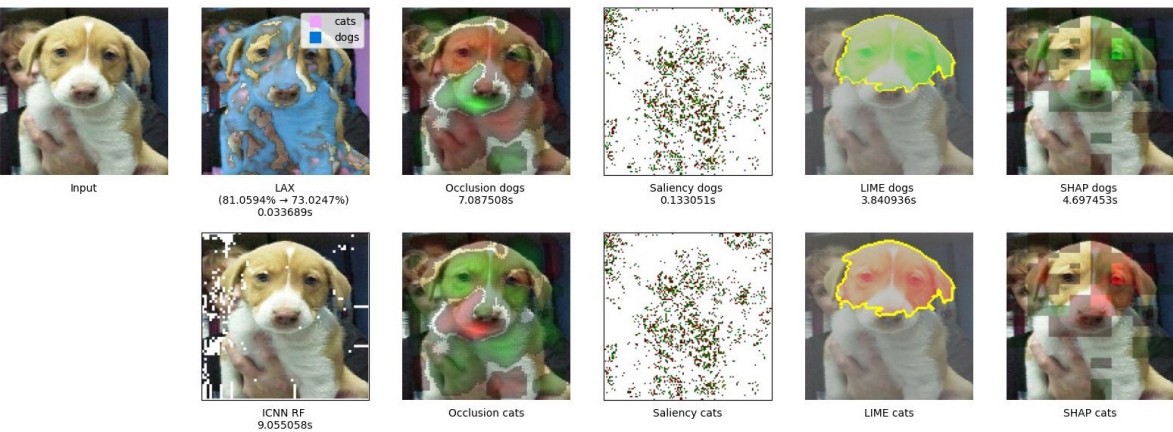

(c) A not-so-successful occlusion example that seems to identify the ground truth signal well.

Figure 4: Additional examples of the LAX method on the Cats and Dogs task. The first column contains the input image. The second column contains the output of the LAX model and the 'Receptive Field' of the Interpretable DenseNet. The last four columns are XAI methods (Occlusion Sensitivity, Saliency Maps, LIME, SHAP in that order) applied to DenseNet. Time in seconds to produce the explanation for the given instance is included for each method.

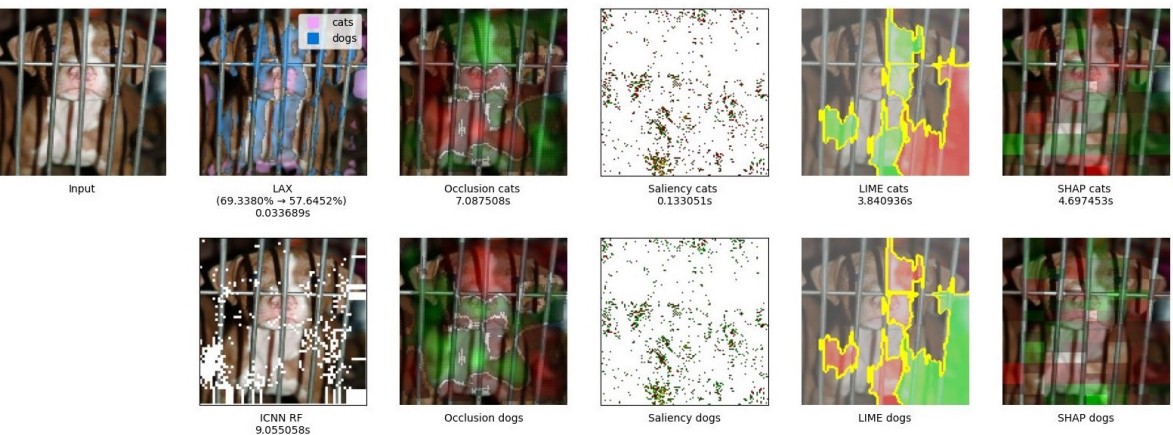

(a) A nearly successful occlusion example that seems to identify the ground truth signal well.

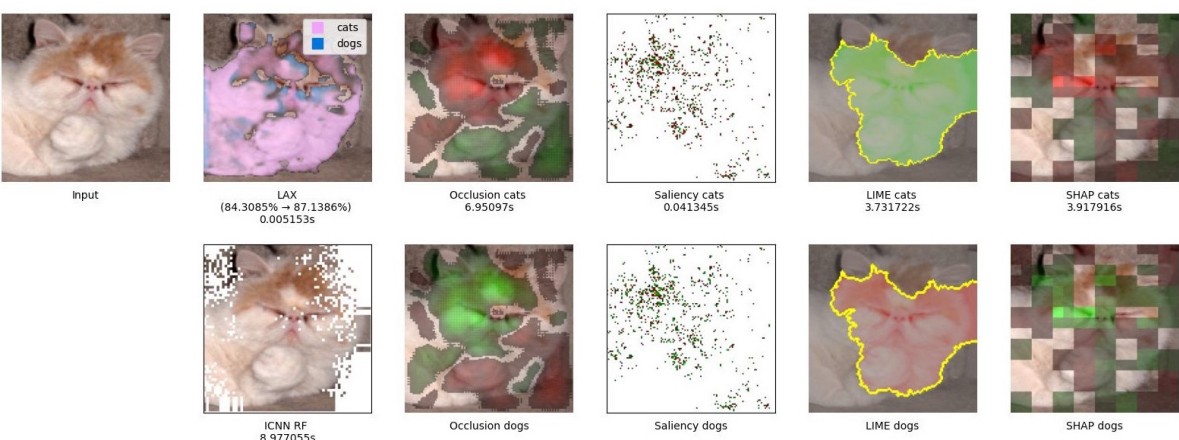

(b) A very unsuccessful occlusion example that seems to identify the ground truth signal well.

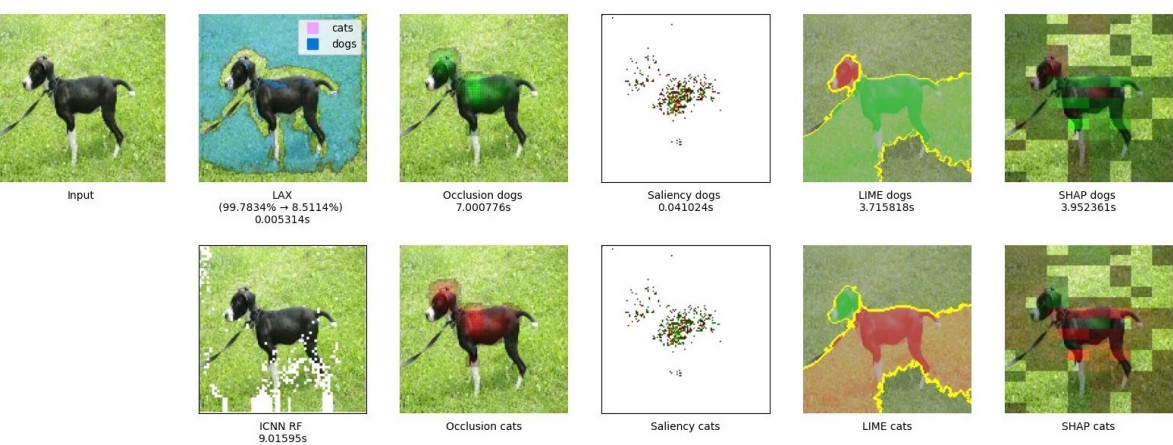

(c) A very successful occlusion example that seems not to identify the ground truth signal at all.

Figure 5: Additional examples of the LAX method on the Cats and Dogs task. The first column contains the input image. The second column contains the output of the LAX model and the 'Receptive Field' of the Interpretable DenseNet. The last four columns are XAI methods (Occlusion Sensitivity, Saliency Maps, LIME, SHAP in that order) applied to DenseNet. Time in seconds to produce the explanation for the given instance is included for each method.

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

## A    Hyperparameter Configuration

The relevant hyperparameters for training are included in Table 5. These hyperparameters were selected through a grid search of the combinations of $\alpha$ and $\beta$ to find the combinations which achieve the highest prediction accuracy and best occlusions. The other optimization parameters were selected to achieve a stable training trajectory using a single NVIDIA A100 GPU. We direct the reader to the code repository for this paper for further details on model optimization.

Table 5: Hyperparmeter configuration for highest accuracy and best occluding models.

| config | Highest Accuracy | Best Occluding |
|---|---|---|
| optimizer | Adam | Adam |
| optimizer momentum | $\beta_1, \beta_2 = 0.9, 0.999$ | $\beta_1, \beta_2 = 0.9, 0.999$ |
| $\alpha$ | $2^{-13}$ | $2^{-10}$ |
| $\beta$ | $2^{-13}$ | $2^{-10}$ |
| base learning rate | 5e-4 | 5e-4 |
| batch size | 97 | 97 |

