# OpenReview forum: "Localized Additive Explanations (LAX)"
_TMLR — Decision pending for TMLR_

### Review · Reviewer_zpyV · 2026-03-27

**Summary Of Contributions:**

The paper proposes Localized Additive Explanations (LAX), an intrinsically interpretable image-to-image neural network architecture designed to address the high computational cost and unreliability of post-hoc Explainable AI (XAI) methods like LIME and SHAP. Instead of using global pooling to output a flat probability vector, LAX outputs a spatial tensor of size $ W \times H \times (k+1) $, representing class-specific masks (plus one background channel). To obtain the final class prediction, the method computes the spatial mean of each class mask and renormalizes these means after excluding the background channel. To train this without ground-truth mask annotations, the authors introduce a clever three-part self-supervised masking loss: $ \ell_0 $ for standard prediction, $ \ell_1 $ to penalize the network if occluding the predicted mask does not drop the confidence, and $ \ell_2 $ forcing a uniform distribution when all class masks are occluded.

**Audience:**

Yes

**Audience Explanation:**

*    By generating the explanation (mask) natively during a single forward pass, LAX achieves inference speeds orders of magnitude faster than perturbation-based methods (e.g., $ 0.002 $ mins vs. $ \sim 2.0 $ mins for SHAP).
*    The three-stage self-contradictory/occlusion training paradigm is intuitively sound. Forcing the model to output a uniform distribution (maximum entropy) when all learned features are masked out is a theoretically solid way to prevent shortcut learning on background textures.
*   Unlike LIME or SHAP, which rely on random perturbations and superpixel segmentation (making them hyperparameter-sensitive and stochastic), LAX provides deterministic and stable explanations for a given input.

**Claims And Evidence:**

No

**Claims Explanation:**

*  The model is only evaluated on a perfectly balanced, binary classification task (Cats vs. Dogs, where $ k=2 $). The architecture requires outputting a spatial tensor of $ W \times H \times (k+1) $. If applied to standard benchmarks like ImageNet ($ k=1000 $), forcing a spatial Softmax across 1001 channels will cause severe training dynamics issues. The gradients for the vast majority of empty classes will vanish, making the network extremely difficult to optimize. Furthermore, performing the 3-stage masking forward passes on a 1001-channel spatial tensor significantly degrades training efficiency compared to standard classification heads.
*   To compute the final probability vector, LAX averages the spatial masks and renormalizes them *after excluding the "no class" background channel*. While this resolves the issue of small objects, it introduces a severe risk of spurious high confidence. If an image contains pure background (neither cat nor dog), the network might output negligible but non-zero noise values for both classes (e.g., Cat: $ 0.002 $, Dog: $ 0.001 $). By discarding the background channel and renormalizing, the network is forced to output a highly confident prediction (e.g., $ \sim 66\% $ Cat), despite lacking any meaningful evidence. This directly contradicts the goal of a trustworthy XAI model.
*    In real-world long-tailed datasets, the network is naturally biased to produce higher initial logits for majority (head) classes. Due to the exponentiation in the pixel-level Softmax, this bias will exponentially suppress the mask activations of minority (tail) classes to near zero. Consequently, tail-class features will likely be erroneously absorbed by the background channel ($ k+1 $). LAX lacks a mechanism to prevent mask collapse for underrepresented classes.
*   Table 2 shows a significant accuracy drop (from $ 89.77\% $ in the DenseNet baseline to $ 84.89\% $ in LAX). Furthermore, the paper lacks a systematic ablation study on the loss weights $ \alpha $ and $ \beta $ to demonstrate how sensitive the mask generation is to these hyperparameters.

**Requested Changes:**

1.  Please provide experimental validation on a dataset with a larger number of classes (e.g., CIFAR-100 or a subset of ImageNet). It would be highly beneficial to discuss or demonstrate how the framework manages optimization stability and Softmax gradient dilution when $ k \geq 100 $.

2.  Consider adding experiments or theoretical discussions to clarify how the model avoids producing falsely confident predictions on Out-of-Distribution (OOD) images or pure background images, given the background-exclusion normalization mechanism.

3.  It is suggested to evaluate the framework's robustness on a long-tailed dataset. This would help verify whether the pixel-level Softmax mechanism maintains adequate mask activations for minority classes without them being overshadowed by majority classes.

4.  The manuscript would benefit from a more comprehensive ablation study detailing the impact of the loss weights $ \alpha $ and $ \beta $. Specifically, visualizing the masks to check for pathological behaviors (e.g., background bleeding or area collapse) when the uniform distribution constraint ($ \ell_2 $) is removed (i.e., $ \beta = 0 $) would provide deeper insights into the necessity of the loss formulation.

5.  Please investigate whether the sharp artificial boundaries created by filling masked regions with mean pixel values introduce high-frequency artifacts. It is important to confirm that the network is learning true semantic features rather than exploiting these edge artifacts as a shortcut to minimize the occlusion losses.

---

> ### Author Response · Authors · 2026-05-09
> **Response to Revisions Requested by Reviewer zpyV**
>
> Reviewer zpyV:
>
> We thank the reviewer for their thorough and insightful review and address each of the requested revisions below.
>
> **Revision 1: Experiments on a dataset with many classes.**
>
> We will conduct additional experiments on CIFAR-100 to explore how our method scales with a large number of classes.  In addition to the impact it may have on optimizing models produced using LAX in terms of occlusion performance and accuracy, we will also examine how the per-class explanation masks behave as the number of competing classes grows.
>
> **Revision 2: Behavior on out-of-distribution and pure background images.**
>
> We would like to clarify that if a LAX model is predicting confidently on an OOD example because it is attributing some part of the input image to an erroneous class this is not a failure of LAX.  We still consider the explanation truthful as long as when that region of the image is masked the prediction changes.  We do, however, agree that it would be informative to investigate such OOD cases to evaluate how LAX performs under these conditions.  Furthermore we believe the case of background images is very important, and we will include experiments to evaluate this case as well.
>
> **Revision 3: Robustness to class-imbalance.**
>
> We agree that class-imbalance is an important consideration for image classification, however we leave that particular case to future work.  There are many different ways of mitigating poor performance under class-imbalance including optimization tricks, self-supervised pre-training, and dataset manipulation.  We do not believe it is in-scope for this particular work to provide a comprehensive treatment of methods we could use to combat class-imbalance in this case, though we do agree it is a valid concern that is relevant to many real-world applications.
>
> **Revision 4: Ablation study on $\alpha$ and $\beta$.**
>
> We will include an ablation study of $\alpha$ and $\beta$ to show the sensitivity of LAX to the change in these parameters.
>
> **Revision 5: Shortcut learning from artificially sharp mask boundaries.**
>
> This is an important issue to address.  To verify that LAX is identifying semantically-relevant features to its prediction rather than taking shortcuts at later steps we will empirically evaluate the performance of LAX under random masking.  We thank the reviewer for this suggestion as we believe it will strengthen the validity of our claims regarding the faithfulness of LAX explanations.
>
> We are grateful for the reviewer's careful and constructive feedback, which we believe will substantially improve the revised manuscript.

---

> > ### Comment · Reviewer_zpyV · 2026-06-02
> >
> > Thank the authors for the rebuttal and the additional experiments.
> >
> > The planned additions (e.g., CIFAR-100, ablations, and random masking) are appreciated and partially address my empirical concerns. However, my core architectural concerns remain unresolved, as they stem from fundamental mathematical mechanisms in LAX rather than generic evaluation metrics:
> >
> > 1. The issue with OOD or pure background images is not about the "truthfulness" of the mask, but a structural flaw. By discarding the background channel and renormalizing, microscopic noise values are mathematically forced to sum to 1.0. The architecture inherently loses the ability to express uncertainty or output low confidence, artificially forcing falsely confident predictions.
> >
> > 2. While class imbalance is a broad topic, my concern is specific to the LAX architecture. The spatial Softmax forces pixel-level competition. This means the naturally lower logits of tail classes will be exponentially suppressed and physically absorbed by the background channel, causing a structural failure to generate masks for minority classes.
> >
> > Because these fundamental architectural limitations remain unaddressed and are not structurally mitigated, I will maintain my current score. I strongly encourage the authors to deeply discuss these mechanistic limitations in the paper.

---

### Review · Reviewer_Wb7k · 2026-03-29

**Summary Of Contributions:**

Authors propose to find out the explanations for deep neural network§s classification decisions.  In order to do so, they design a network architecture for producing per-class masks, and define a custom loss function combining supervised and self-supervised signals. They showed that the method often, but not always, gives meaningful results.

The weaknesses of the paper include
- not recognizing that their approach inherently seeks texture-based explanations. Doing so has some rationale (see e.g. https://openreview.net/pdf?id=Bygh9j09KX). However, recent work suggests that CNNs use not only textures (https://arxiv.org/abs/2509.20234). Discussion of this point is entirely missing from the article and maybe could explain unexpected behavior in Fig. 4.
- I don't think the concept of the mask is properly explained. Mask could be binary (elements from $\{0,1\}$), soft (elements from $[0,1]$ or $(0,1)$), or arbitrary (elements from $\boldsymbol{R}$).  On page 4 of their paper it appears the mask would be the last case, but that would make masks very hard to interpret.
- A weakness seems to be difficulty to generalize to many classes. However, in One-vs-Rest classification, a common way to solve multiclass problems, even the binary case is useful. I would appreciate indication whether the authors tried many class problems and what difficulties they have encountered.

**Additional Comments:**

In my opinion, the second component of the loss function is unstable which would make training process fragile. I would propose replacing it with a more symmetric term - the weighted sum over all classes.

**Audience:**

Yes

**Audience Explanation:**

Explanations of neural network inner workings would be very desirable. The question is whether this is feasible at all, even under the assumption that texture is what guides neural network's decisions. Nevertheless, the method seems to work reasonably well, so this maybe useful for further investigations of the proposed mask-based approach.

**Broader Impact Concerns:**

I do no see any broader impact concerns.

**Claims And Evidence:**

Yes

**Claims Explanation:**

The authors' method is readily visualized, and the article contains ample illustrations of successes and failures of the method. The authors provide comparisons with competing methods.

**Requested Changes:**

All three weaknesses listed above should be addressed, either by comments or by changes in the paper.

---

> ### Author Response · Authors · 2026-05-09
> **Response to Revisions Requested by Reviewer Wb7k**
>
> Reviewer Wb7k:
>
> We thank the reviewer for their detailed and constructive review and respond to each requested revision below.
>
> **Revision 1: Texture-based nature of the explanations.**
>
> Our approach is not limited to any single model type such as a CNN-based U-Net which may attend mostly to texture-based features.  We agree that some model types may pick out texture features, but other model architectures such as transformers may not be as sensitive to textures.  We agree that certain model architectures may utilize certain aspects of the input image to make predictions, and the goal of LAX is to identify those.
>
> **Revision 2: Clarification of the mask concept.**
>
> Upon rereading the content in page 4 we agree that we were not as clear as we should have been.  As the masks are slices of the output volume of the network they take values in the interval [0, 1].  They represent, at each pixel location, the probability of that pixel being evidence for a given class or the sum of probabilities for all classes as in the case of $M_{sum}$.  We will make sure to be more precise when stating to which set the masks belong.
>
> **Revision 3: Experiments on a dataset with many classes.**
>
> We designed LAX with the intention that it be used on multi-class classification problems as well as binary classification problems.  We have currently conducted experiments on CIFAR-10 and the LAX models are still able to predict and occlude well.  Having read the other reviewers' comments, we believe it is important to understand the performance of LAX on datasets with a very large number of classes.
>
> We greatly appreciate the reviewer's engagement with our work and look forward to incorporating these revisions into the manuscript.

---

> > ### Comment · Reviewer_Wb7k · 2026-05-09
> > **Clarification on textures**
> >
> > It would be very interesting to see in the paper an example for a LAX based classification network using transformers. I think it would enhance the paper considerably. Absent that, I stand by on my request to discuss textures in more detail.

---

> > ### Comment · Reviewer_Wb7k · 2026-05-09
> > **CIFAR-10**
> >
> > CIFAR-10 is often used in vision classification demonstrations. Personally, I view the images as too small. Maybe ImageWoof subset of Imagenet would be a better fit.

---

### Review · Reviewer_eLfS · 2026-04-26

**Summary Of Contributions:**

Localized Additive eXplanations (LAX) is an interesting idea. The goal is to produce occlusions to identify which image regions are essential for a given classification. This is not novel, but making these additive would be relevant. The approach is to use a custom loss function to make explanations exclusive (covering nothing but evidence) as well as correct and complete (although there is a problem with this, see below). The main claim is that compared with LIME, SHAP, and other occlusion methods, LAX is several orders of magnitude faster while achieving competitive accuracy, trading a small amount of classification performance for built-in explainability.

**Additional Comments:**

I'd encourage the authors to revise and expand the empirical evaluation of the paper considerably, and then consider making a re-submission because the broader XAI problem has not been solved entirely and research in this area is highly needed.

**Audience:**

Yes

**Audience Explanation:**

This line of research continues to be extremely relevant and significant to the TMLR audience. It is closely connected with recent efforts to enhance reasoning capabilities and to evaluate reasoning in neural networks. Whether the underlying network is a CNN, ViT or GAN there are now many real applications of these models, in particular in medicine, that require reliable local XAI technology.

**Broader Impact Concerns:**

There are no broader concerns. If anything, research in this area (XAI) serves to address some of the main existing concerns. XAI is a relevant technology in the efforts to achieve accountability in AI.

**Claims And Evidence:**

No

**Claims Explanation:**

There are two main issues that need addressing: (1) claims of correctness and completeness are not backed up in the paper by formal results or sufficient empirical evaluation; (2) the novelty of engineering the loss function to achieve essentially contrastive counterfactual explainability is low (see recent surveys on XAI, e.g. https://www.scopus.com/pages/publications/85202175755?origin=resultslist&source=sd-apx).

Another important aspect is that LIME and SHAP are no longer adequate as a baseline for comparison. The problems with these methods is now very well known and vastly documented. So the baseline for comparison should be more closely-related work mentioned.

Finally, the experimental evaluation needs to be extended. It is insufficient to present a few examples of explainability without a measure of XAI fidelity. This is a recurring problem of local XAI. As far as I know, the first paper to introduce a fidelity measure for local XAI (and in the process also showing the serious problems with LIME) was this: https://arxiv.org/abs/1908.03020.

The comparative empirical evaluation, instead of being carried out on a few isolated examples that only serve to illustrate specific cases shown in the paper, should be done systematically using existing benchmarks for comparison. This work may help you choose the benchmarks and metrics: https://arxiv.org/abs/2106.14556, as well as discuss what would happen with your method in the important case of overdetermination.

**Requested Changes:**

Comparative evaluations should be carried out using the same benchmarks as used by the closest related work, not LIME and SHAP, that is work on counterfactual explainability for images with overdetermination, measured using a local XAI fidelity metric.

The claim that no post-hoc XAI is required should be evaluated taking into account any possible negative effect of imposing additive explainability, e.g. is there a corresponding drop in accuracy when the model is additive?

The claims of correctness and completeness should be either proved or toned down: "LAX models are trained to optimize objectives which ensure that the signals identified by the explanation are ‘correct’ (they are evidence for a class), ‘complete’ (essential evidence for predicting a class), and ‘exclusive’ (not identifying anything other than evidence for a class)." In the paper's current form, these claims are not supported.

---

> ### Author Response · Authors · 2026-05-09
> **Response to Revisions Requested by Reviewer eLfS**
>
> Reviewer eLfS:
>
> We thank the reviewer for their careful examination of our work and address each of the requested revisions below.
>
> **Revision 1: Counterfactual comparisons with local XAI fidelity metric.**
>
> We thank the reviewer for bringing these references to our attention.  They are certainly highly relevant related work, and we agree that our evaluation could be strengthened by comparing to these methods for counterfactual explainability.  We agree that the addition of a fidelity metric and benchmark comparisons would increase the strength of our evaluation as well.
>
> We find the notion of over-determination important, and we will make sure to include a discussion of the performance of LAX in the case of over-determined predictions, where multiple disjoint regions of the input may each be sufficient to support the predicted class.
>
> **Revision 2: Limitations section addressing the accuracy trade-off.**
>
> We do address the fact that our method sacrifices some accuracy on the cats and dogs task we evaluate it on. We will add a limitations section to our paper to address this more comprehensively, as well as the other limitations of our method such as architecture-dependence.
>
> **Revision 3: Justification of correctness, exclusivity, and completeness claims.**
>
> We believe that our notion of the correctness of an explanation is supported in the text by defining a correct explanation to be one that changes the class label.  We do agree that we could better substantiate claims of completeness and exclusivity, however.
>
> We thank the reviewer once again for their thoughtful comments, which we believe will improve the quality of our manuscript.